# Mental and somatic disorders and the subsequent risk of all-cause and cause-specific mortality in refugees, non-refugee migrants and the Swedish-born youth: a population-based cohort study in Sweden

Magnus Helgesson [1], Emma Björkenstam,[1,2] Svetlana Filatova,[1] Syed Ghulam Rahman,[1] Alexis Cullen,[1,3] Thomas Dorner,[1,4] Katalin Gémes [1], Ridwanul Amin [5], Ellenor Mittendorfer-Rutz [1]

For numbered affiliations see end of article.

**Correspondence to**
Dr Ellenor Mittendorfer-Rutz;
ellenor.mittendorfer-rutz@ki.se

## ABSTRACT

**Objectives** The aims were to investigate the associations between specific mental and somatic disorders and subsequent all-cause and cause-specific mortality (suicide, external and other causes) in young refugees and non-refugee migrants compared with Swedish-born individuals of similar age.

**Methods** In this register-based prospective cohort study, all 1 003 760 individuals (40 305 refugees, 31 687 non-refugee migrants as the exposure groups and the rest as the Swedish-born comparison group), 16–25 years old, residing in Sweden on 31 December 2004 were included. These individuals were followed regarding the outcome of all-cause and cause-specific mortality (suicide and external causes) between 2005 and 2016. The study population was also stratified according to any use of specialised healthcare for mental or somatic diagnoses before baseline (2000–2004). Cox regression models yielding crude and multivariate Hazard Ratios (HR and aHR, respectively) with 95% Confidence Intervals (CI) were used to investigate the afore-mentioned associations.

**Results** A lower proportion of both refugees (12%) and non-refugee migrants (10%) had college/university education compared with the Swedish-born individuals (17%). The proportion of unemployed (>180 days) among refugees (2.3%) and non-refugees (2.9%) was higher than the Swedish born (1.4%). Refugees and non-refugee migrants had about a 20% lower risk of all-cause mortality and external causes of mortality compared with Swedish-born individuals. An even greater reduction in suicide risk (aHR 0.51, 95% CI 0.37 to 0.70, and 0.63, 95% CI 0.49 to 0.82 for non-refugees and refugees, respectively) was found. When restricted to those with a mental or somatic disorder, a lower risk of both general and specific mortality was also found among both refugees and non-refugee migrants compared with Swedish-born individuals. Refugees had, however, equal point estimates of all-cause mortality associated with substance misuse disorder and neoplasms as their Swedish-born peers with these disorders.

## STRENGTHS AND LIMITATIONS OF THIS STUDY

⇒ Longitudinal population-based design and use of national registers with high completeness and validity.
⇒ Lack of data from primary healthcare may have led to differential underestimation of the mental and somatic disorders among migrant groups than Swedish-born host population.
⇒ Generalisability is limited to countries with similar healthcare and social welfare systems, and populations with similar ethnic composition compared with Sweden.
⇒ The results cannot be generalised to individuals living in refugee camps or immigration detention centres.

**Conclusions** With few exceptions, young migrants with specific mental and somatic disorders have a mortality advantage compared with their Swedish-born peers with the same disorders. Further research on protective factors is warranted.

## INTRODUCTION

During the past few years, there has been a substantial increase in global migration. In 2017, there were 258 million migrants worldwide, including 25.4 million refugees.[1 2] Sweden is one of the largest European recipients of refugees, during the last two decades, more than three quarters of a million refugees sought asylum in Sweden.[3] About one-third of them were children, and a large proportion arrived in Sweden unaccompanied by parents or relatives. This development has the potential to be an eminent public health challenge, as many of the young refugees have

experienced traumatic events, such as bereavement and violence, during war.[3–5] For these reasons, studies report a higher risk for mental disorders, particularly post-traumatic stress disorders in young refugees compared with the host population.[6–12]

Despite the large numbers of young refugees world-wide and their higher morbidity risk, there is a scarcity of studies focussing on the most prominent health outcomes, that is, mortality, in this group. Mortality is a relatively rare event among young persons in general and death are mainly attributable to external causes including suicide, injuries, accidents and poisoning.[13–15] Studies investigating mortality across all age groups have reported a lower mortality risk in refugees compared with the host population.[16 17] This migrant mortality advantage has puzzled researchers as it occurs against a background of higher morbidity levels in these populations.[18] Explanatory models include not only in-migration and out-migration health selection, but also cultural features such as better health behaviour but also data artefacts, primarily due to misclassification of age, nationality and ethnicity.[16 18 19] There seems to be an age variation in this migrant mortality advantage, however, where protection from excess mortality does not appear to apply to children as it does among their older counterparts.[18] Young refugees might be at a particular mortality risk, in part, as their main causes of death, suicide and external causes are strongly related to mental disorders, which are known to have an early age of onset and high prevalence rates in young refugees.[20–22] It is, therefore, crucial to investigate the mortality risk in young refugees.

Besides these higher prevalence rates of mental disorders, the occurrence of some somatic disorders (any non-psychiatric disorder), including, for example, infectious diseases as well as diseases of the respiratory system and the nervous system, appears to be prevalent in refugees.[17 23] Variation in the importance of specific mental disorders between adult refugees and Swedish born regarding the subsequent risk of suicide mortality has also been recently reported.[15] Therefore, specific mental and somatic disorders may play a differential role for the mortality risk among young refugees compared with young individuals in the host population. However, such knowledge is lacking to date. Any such differences might arise from unmet healthcare and treatment needs of young refugees with specific disorders.[24] Here, adequate care in young refugees might be hampered by language barriers, differences in the clinical manifestation and the course of the underlying disease, as well as the lack of competence in transcultural medicine among healthcare staff in the host country.

Studies report differences in the prevalence of both mental and somatic disorders as well as multimorbidity between subgroups of migrants.[25 26] Refugees have a higher risk of having experienced traumatic events, which might put them at a higher risk of developing different disorders and subsequently premature mortality compared with other migrants.[27] On the other hand, refugees and non-refugee migrants from the same countries of birth might share common culturally determined features, including the perception of healthcare, acceptance of treatment and health behaviour. To be able to create tailor-made interventions, there is, therefore, a need to study different subgroups of migrants separately.

This study aims to address the afore-mentioned gaps in the literature by investigating if the risk of all-cause and cause-specific mortality differs in young refugees and non-refugee migrants compared with individuals in the host population of the same age. A further aim was to investigate if the potential differences in all-cause and cause-specific mortality vary by a prior diagnosis of specific mental and somatic disorders.

## METHODS
### Study population
A cohort study was conducted where the study population was defined as all individuals aged 16–25 years, residing in Sweden on 31 December 2004, that is, born between 1979 and 1988 (n=1 081 412). Only those with complete information on their reason for settlement in Sweden were included (n=1 048 762) because determining refugee versus non-refugee migrant status depends on the availability of this information. Individuals with incomplete information regarding this variable are primarily EU citizens who do not register their (and their families') right of residence with the Swedish Tax Agency. For the non-refugee migrants, we excluded those who did not come from the same countries as the refugees (n=9008) to investigate if traumatic experience as a refugee had affected mortality outcomes given that the postmigration conditions and the cultural background can be considered similar between refugees and non-refugee migrants from the same countries. A further 35 994 individuals were excluded because they did not reside in Sweden during the period 2000–2004, that is, the period during which mental and somatic disorders were measured. The final study population comprised of 1 003 760 young individuals aged 16–25 years, including 40 305 refugees and 31 687 non-refugee migrants with residence permit (temporary residents for <365 days, asylum seekers and undocumented migrants do not have a personal identity number in Sweden, and therefore, their information is not available in the administrative registers). Individuals in the study population were then followed from 2005 to 2016 regarding the outcome measures (please see the section Outcome measures) by linking several nationwide registers.

### Patient and public involvement
As this study builds on administrative registers, patients were not involved in the conduct of this study.

### Registers
We used the unique (deidentified) Swedish personal identity number to link information from several

population-based registers. The researchers had full access to these databases. The Longitudinal Integration Database for Health Insurance and Labour Market Studies contains data on sociodemographic and labour market factors. The Longitudinal Database for Integration Studies holds migration-related information, including reasons for settlement (eg, refugee status). The National Patient Register (NPR) includes information on inpatient healthcare since 1987, and for specialised out-patient care since 2001. Diagnoses in NPR are coded according to the International Classification of Diseases, version 10 (ICD-10). The Cause of Death Register comprises information on causes and date of deaths of Swedish residents since 1952 (and including deaths registered abroad since 2012).

## Measures
### Mental disorders
Mental disorders were defined as having a main diagnosis from specialised healthcare (either inpatient or specialised outpatient healthcare), as recorded in the NPR between 2000 and 2004, of the following disorders: substance misuse disorders (ICD-10: F10–19), schizophrenia/non-affective psychotic disorders (F20–29), depressive disorders (ICD-10: F32–34), anxiety disorders (ICD-10: F40–42), stress-related disorders (F43) and behavioural disorders (ICD-10: F50–59, F90–99).

### Somatic disorders
Main diagnoses in specialised healthcare (2000–2004) for somatic disorders were divided into dichotomised variables of following groups of diagnoses, recorded according to the ICD-10: (1) infectious and parasitic diseases (ICD-10: A00–B99), (2) neoplasm (ICD-10: C00–D48), (3) diseases of the circulatory system (ICD-10: I00–I99), (4) diseases of the respiratory system (ICD-10: J00–J99), (5) diseases of the digestive system (ICD-10: K00–K93), (6) musculoskeletal diseases (ICD-10: M00–M99), (7) diseases of the nervous system (ICD-10: G00–G99) and (8) other somatic disorders (ICD-10: E00–E90, H01–H99, L00–L99, N00–T99).

### Outcome measures
The outcome measures were defined as both all-cause and cause-specific mortality. More specifically, we studied external causes (ICD-10: V00–Y99, including both intentional and unintentional injuries) as one outcome, and within this group, suicide (ICD-10: X60–84, Y10–34) was examined separately.

### Covariates
The covariates included in the analyses were (1) sociodemographic factors: sex, age, educational level, type of living area and family situation (all variables were measured at baseline, ie, 31 December 2004); (2) work-related factors: unemployment (0, 1–180, >180 days); sickness absence (0, 1–90, >90 days) and disability pension, all measured during 2004. Missing values in any covariates were treated as a separate category in the

adjusted analyses. Categorisations of covariates are shown in table 1.

## Statistical analyses
Statistical analyses were conducted using SAS, V.9.4. Crude and multivariate analyses were performed using Cox regression models of time to diagnosis-specific and all-cause mortality during the follow-up. The proportional hazard assumption was verified graphically using log-minus-log plots (plotting the log–log transformation of the survival function against survival time) over the different categories for each of the covariates and exposure variables. We assessed person-years at risk by totalling the years that the individuals were living in Sweden during the follow-up period. Calendar day was used as the underlying timescale and age at study entry was adjusted for in the multivariate-adjusted models. The entry date was defined as 1 January 2005, and the exit date as the date of the first outcome, date of emigration or the end of follow-up on 31 December 2016. All individuals who emigrate are expected to notify the Swedish Tax Agency and this information is updated in all administrative registers. However, as it is possible that not everyone reports the emigration status, we additionally identified all individuals who did not have any information in the population register for two consecutive years and considered that they have emigrated during the former year (if they had not died before that) to compensate for the under-reporting. First, risk estimates for refugees and non-refugee migrants were compared with those for Swedish born regarding subsequent all-cause mortality and cause-specific mortality. The association of mental and somatic disorders and subsequent all-cause mortality with refugee status was then assessed. Here, analyses were performed for any mental or somatic disorders and then we conducted separate analyses for each diagnostic group (only specific disorders where number of cases exceeded 10 either in refugees or non-refugee migrants are shown in tables). We examined the associations between migrant status and each outcome in one crude and one adjusted regression model. Model 2 was adjusted for age, sex, education, family situation, type of living area, unemployment, sickness absence and disability pension, and finally morbidity (ie, mental and somatic disorders as described above). Due to lack of data on time-dependent variables, we measured these covariates only at baseline. All covariates known to be associated with both the exposure and the outcome that do not fall in the causal pathway and for which data were available were included in the multivariate adjusted models. The directed acyclic graphs of unadjusted and adjusted models (please see online supplemental material) did not reveal any biased pathway due to colliders. Moreover, multicollinearity among the covariates was tested using Cramér's V. Analyses investigating the association with mental disorders and subsequent mortality were adjusted for somatic comorbidity and vice versa. The interaction between refugee status and mental/somatic disorders was

**Table 1** Cohort and health-related characteristics, by refugee status, in individuals aged 16–25 years old residing in Sweden in 2000–2004

| | All | Swedish-born individuals n (column %) | Refugees n (column %) | Non-refugee migrants n (column %) |
|---|---|---|---|---|
| All, (n, row per cent) | 1 003 760 (100) | 931 768 (93) | 40 305 (4) | 31 687 (3) |
| **Sociodemographic factors at baseline** | | | | |
| Sex | | | | |
| Women | 487 254 (49) | 452 142 (49) | 19 041 (47) | 16 071 (51) |
| Men | 516 506 (51) | 479 626 (51) | 21 264 (53) | 15 616 (49) |
| Age group | | | | |
| 16–20 | 525 724 (52) | 487 676 (52) | 21 378 (53) | 16 670 (53) |
| 21–25 | 478 036 (48) | 444 092 (48) | 18 927 (47) | 15 017 (47) |
| Mean age (years, SD) | 20.3 (2.9) | 20.3 (2.9) | 20.3 (2.8) | 20.4 (2.9) |
| Education (years) | | | | |
| Compulsory school (<9) | 408 595 (41) | 374 906 (40) | 18 354 (46) | 15 335 (48) |
| High school (10–12) | 407 911 (41) | 383 312 (41) | 14 848 (37) | 9751 (31) |
| College or university (>12) | 162 977 (16) | 154 907 (17) | 4994 (12) | 3076 (10) |
| Missing | 24 277 (2) | 18 643 (2) | 2109 (5) | 3525 (11) |
| Family situation | | | | |
| Married/living with partner without children* | 8915 (1) | 6342 (1) | 1518 (4) | 1055 (3) |
| Married/living with partner with children* | 33 653 (3) | 28 629 (3) | 2371 (6) | 2653 (8) |
| Single/divorced/separated/widowed without children* | 483 651 (48) | 453 573 (49) | 17 067 (42) | 13 011 (41) |
| Single/divorced/separated/widowed with children* | 9694 (1) | 8437 (1) | 608 (2) | 649 (2) |
| Children (≤20 years old)[2] | 467 847 (47) | 434 787 (47) | 18 741 (46) | 14 319 (45) |
| Type of living area | | | | |
| Big city area | 349 066 (35) | 313 513 (34) | 17 908 (44) | 17 645 (56) |
| Intermediate (>90 000 inhabitants) | 381 280 (38) | 355 443 (38) | 16 121 (40) | 9716 (31) |
| Small (rural municipalities) | 273 414 (27) | 262 812 (28) | 6276 (16) | 4326 (14)) |
| Country/region of birth | | | | |
| Horn of Africa† | 4675 (0) | – | 1917 (5) | 2758 (9) |
| Afghanistan | 1124 (0) | – | 329 (1) | 795 (3) |
| Iraq | 8464 (1) | – | 4120 (10) | 4344 (14) |
| Iran | 7366 (1) | – | 4773 (12) | 2593 (8) |
| Syria | 1942 (0) | – | 1423 (4) | 519 (2) |
| Other countries from Asia (except Afghanistan, Iraq, Iran and Syria) | 12 645 (1) | – | 4631 (11) | 8014 (25) |
| Former Yugoslavian countries | 19 576 (2) | – | 17 852 (44) | 1724 (5) |
| Other countries (except Horn of Africa, Asian countries and former Yugoslavian countries)‡ | 16 200 (2) | – | 5260 (13) | 10 940 (35) |
| Length of formal residency in Sweden | | | | |
| Mean number of years (SD) | | – | 11.3 (3.4) | 10.2 (4.4) |
| 0–4 years | 4975 (0) | – | 1742 (4) | 3233 (10) |
| 5–10 years | 28 887 (3) | – | 15 527 (39) | 13 360 (42) |
| >10 years | 37 947 (4) | – | 22 923 (57) | 15 024 (47) |
| Missing | 183 (0) | – | 113 (0) | 70 (0) |
| **Work-related factors at baseline** | | | | |
| No unemployment | 806 138 (80.3) | 752 466 (80.8) | 29 452 (73.1) | 24 220 (76.4) |

Continued

**Table 1** Continued

| | All | Swedish-born individuals n (column %) | Refugees n (column %) | Non-refugee migrants n (column %) |
|---|---|---|---|---|
| Unemployment≤180 days/year | 182 263 (18.2) | 165 847 (17.8) | 9670 (24.0) | 6746 (21.3) |
| Unemployment>180 days/year | 15 359 (1.5) | 13 455 (1.4) | 1183 (2.9) | 721 (2.3) |
| No sickness absence (SA) | 968 476 (96.5) | 898 851 (96.5) | 38 822 (96.3) | 30 803 (97.2) |
| SA≤90 days/year | 25 798 (2.6) | 24 078 (2.6) | 1094 (2.7) | 626 (2.0) |
| SA>90 days/year | 9486 (0.9) | 8839 (0.9) | 389 (1.0) | 258 (0.8) |
| No disability pension (DP) | 989 698 (98.6) | 918 659 (98.6) | 39 789 (98.7) | 31 250 (98.6) |
| DP | 13 762 (1.4) | 12 809 (1.4) | 516 (1.3) | 437 (1.4) |
| **Disorders at baseline** | | | | |
| Mental disorders§ | | | | |
| Any mental disorder | 48 564 (4.8) | 45 004 (4.8) | 1803 (4.5) | 1757 (5.5) |
| Substance misuse disorders | 15 848 (1.6) | 14 752 (1.6) | 552 (1.4) | 544 (1.7) |
| Schizophrenia/non-affective psychotic disorders | 1853 (0.2) | 1590 (0.2) | 128 (0.3) | 135 (0.4) |
| Depressive disorders | 11 232 (1.1) | 10 588 (1.1) | 321 (0.8) | 323 (1.0) |
| Anxiety disorders | 9040 (0.9) | 8520 (0.9) | 286 (0.7) | 234 (0.7) |
| Stress-related mental disorders | 5598 (0.6) | 4938 (0.55) | 347 (0.9) | 313 (1.0) |
| Behavioural disorders | 9016 (0.9) | 8492 (0.9) | 232 (0.6) | 292 (0.9) |
| Somatic disorders§ | | | | |
| Any somatic disorder | 516 246 (51.4) | 477 497 (51.2) | 21 547 (53.5) | 17 202 (54.3) |
| Infectious and parasitic diseases | 39 940 (4.0) | 36 433 (3.9) | 1922 (4.8) | 1585 (5.0) |
| Neoplasm | 29 669 (3.0) | 28 179 (3.0) | 847 (2.1) | 643 (2.0) |
| Diseases of the circulatory system | 10 028 (1.0) | 9282 (1.0) | 417 (1.0) | 329 (1.0) |
| Diseases of the respiratory system | 66 763 (6.7) | 62 347 (6.7) | 2462 (6.1) | 1954 (6.2) |
| Diseases of the digestive system | 52 144 (5.2) | 48 135 (5.2) | 2299 (5.7) | 1710 (5.4) |
| Musculoskeletal diseases | 65 865 (6.6) | 61 347 (6.6) | 2530 (6.3) | 1988 (6.3) |
| Diseases of the nervous system | 17 913 (1.8) | 16 581 (1.8) | 740 (1.8) | 592 (1.9) |
| Other somatic diseases | 419 249 (41.8) | 386 602 (41.5) | 18 028 (44.7) | 14 619 (46.1) |

*Living at home.
†Somalia, Eritrea and Ethiopia.
‡Other countries include primarily South American and remaining African countries.
§During 2000–2004.

investigated by stratified analyses. Results are presented as crude and adjusted Hazard Ratios (HRs and aHRs) with 95% Confidence Intervals (CI).

## RESULTS

Among refugees, the percentage of men (53%) was found comparable to both Swedish-born (51%) and non-refugee migrants (49%, table 1). There was a higher percentage of Swedish-born individuals (17%) with college/university education compared with both refugees (12%) and non-refugee migrants (10%). Among the refugees, the majority came from Former Yugoslavian countries (44%), whereas non-refugee migrants primarily emigrated from Asian countries. Most of the migrants had lived in Sweden for more than 10 years (47%–57%). The percentage of

individuals who were unemployed>180 days was more than double among migrants (2.3%–2.9%) compared with Swedish born (1.4%), yet there were no differences between the groups regarding the prevalence of sickness absence and granting of disability pension. The prevalence of any specialised healthcare use due to mental disorder was highest among non-refugee migrants (5.5%) and lowest among refugees (4.5%). Likewise, specialised heatlh care use due to a somatic disorder occurred most frequently in non-refugee migrants (54.5%) followed by refugees (53.5%) and Swedish-born individuals (51.2%).

Both refugees and non-refugee migrants had about a 20% lower risk of all-cause mortality and mortality due to external causes, while the risk of suicide was around 50% lower among non-refugee migrants and nearly 40%

**Table 2** Risk of all-cause and cause-specific mortality in Swedish-born individuals, refugees and non-refugee migrants, aged 16–25 years old residing in Sweden in 2000–2004. HRs with 95% CIs

| | N (rate per 100 000 person-years) | Model 1* | Model 2† |
|---|---|---|---|
| | All-cause mortality | | |
| Swedish-born individuals | 5834 (53.4) | 1 (REF) | 1 (REF) |
| Non-refugee migrants | 186 (52.1) | 0.98 (0.85 to 1.13) | 0.78 (0.67 to 0.91) |
| Refugees | 229 (49.1) | 0.92 (0.81 to 1.05) | 0.80 (0.70 to 0.91) |
| | Suicide | | |
| Swedish-born individuals | 1977 (18.1) | 1 (REF) | 1 (REF) |
| Non-refugee migrants | 39 (10.9) | 0.61 (0.44 to 0.83) | 0.51 (0.37 to 0.70) |
| Refugees | 60 (12.9) | 0.71 (0.55 to 0.92) | 0.63 (0.49 to 0.82) |
| | External causes | | |
| Swedish-born individuals | 3873 (35.5) | 1 (REF) | 1 (REF) |
| Non-refugee migrants | 123 (34.5) | 0.97 (0.81 to 1.17) | 0.78 (0.65 to 0.94) |
| Refugees | 151 (32.4) | 0.91 (0.78 to 1.08) | 0.78 (0.66 to 0.92) |

*Crude.
†Adjusted for age, sex, education, family situation, type of living area, unemployment, sickness absence, disability pension at baseline and mental and somatic disorders in 2000–2004.

lower among refugees compared with Swedish-born individuals (table 2). No differences between refugees and non-refugee migrants emerged. The estimates in the multivariate models in these analyses were only to a minor extent altered by adjustment for the various covariates. The variable which contributed most to the changed estimates was age. Other variables only contributed marginally.

Compared with Swedish-born youth without any mental disorder, Swedish-born individuals with mental disorders had higher risk of all-cause mortality (aHR: 3.4, 95% CI: 3.2 to 3.6) as well as for death by suicide (aHR: 5.8, 95% CI: 5.2 to 6.5) and external causes (aHR: 5.0, 95% CI: 4.6 to 5.4). These estimates were lower in both non-refugee migrants (aHRs: 2.4, 2.6 and 3.0, respectively) and refugees (aHRs: 2.9, 3.8 and 4.4, respectively; table 3). The same pattern, but less pronounced, was seen among those with somatic disorders, that is, Swedish born with somatic disorders had higher risk for all-cause mortality (aHR: 1.6, 95% CI: 1.5 to 1.7), suicide (aHR: 1.3, 95% CI: 1.2 to 1.4) and death due to external causes (aHR: 1.4, 95% CI: 1.3 to 1.4) compared with Swedish born without somatic disorders. These estimates were again lower in migrants: non-refugee migrants (aHRs: 1.3, 0.7 and 1.2, respectively) and refugees (HRs: 1.3, 0.8 and 1.1, respectively).

Similar patterns were also observed for specific mental and somatic disorders with two exceptions: refugees and Swedish-born individuals with substance misuse disorder and neoplasms had equal point estimates of mortality (table 4). Mortality risk for specific disorders in refugees compared with Swedish-born without the specific disorders ranged from 0.9 for infectious and parasitic diseases to 5.3 for substance misuse disorder.

## DISCUSSION
### Main findings
Our study examined the all-cause and cause-specific mortality risk in young refugees and non-refugee migrants with specific mental and somatic disorders. Refugees and non-refugee migrants had, compared with Swedish-born individuals, a 20% lower risk of all-cause mortality and mortality due to external causes. The risk of suicide was even lower (50% lower among non-refugees and nearly 40% lower among refugees) in the multivariate analyses. Similar patterns were found for specific mental and somatic disorders, with the exception of substance misuse disorder and neoplasms, where refugees and Swedish-born individuals had similar risk estimates.

The lower mortality risk in young migrants compared with the host population is in line with previous studies including all age groups.[16 18 28] As reported previously, we could not find any differences in estimates regarding mortality between refugees and non-refugee migrants.[16] This is at odds with the higher occurrence of traumatic experiences and consequently higher risk of mental disorders and worse health status in refugees published previously.[29 30] An explanation for the lack of differences in estimates between refugees and non-refugee migrants may therefore be related to the importance of postmigration factors for subsequent mortality risk. Here, factors such as common cultural features, discrimination and low labour market attachment may affect refugees and non-refugee migrants equally.

Several models have been put forward to explain the mortality advantage among migrants.[16 18] A positive health selection of migrants ('healthy migrant effect') has yielded not only the highest explanatory value, but

**Table 3** Associations between mental and somatic disorders, respectively, and all-cause and cause-specific mortality in Swedish-born individuals and refugee and non-refugee migrants, aged 16–25 years old residing in Sweden in 2000–2004

| Mental/disorder status | Refugee status | n (rate per 100 000 person-years) | Model 1* | Model 2† |
|---|---|---|---|---|
| | | All-cause mortality | | |
| No mental disorder | Swedish born | 4605 (44.3) | 1 (REF) | 1 (REF) |
| | Non-refugee migrants | 149 (44.3) | 1.00 (0.85 to 1.18) | **0.81 (0.68 to 0.95)** |
| | Refugees | 178 (40.0) | 0.90 (0.78 to 1.05) | **0.78 (0.67 to 0.90)** |
| Mental disorder | Swedish-born | 1229 (234.4) | **5.29 (4.97 to 5.64)** | **3.38 (3.16 to 3.62)** |
| | Non-refugee migrants | 37 (183.5) | **4.15 (3.00 to 5.74)** | **2.35 (1.70 to 3.25)** |
| | Refugees | 51 (245.4) | **5.55 (4.21 to 7.31)** | **2.94 (2.23 to 3.88)** |
| | | Suicide | | |
| No mental disorder | Swedish born | 1434 (13.8) | 1 (REF) | 1 (REF) |
| | Non-refugee migrants | 29 (8.6) | **0.63 (0.43 to 0.91)** | **0.54 (0.37 to 0.78)** |
| | Refugees | 44 (9.9) | **0.72 (0.53 to 0.97)** | **0.63 (0.46 to 0.85)** |
| Mental disorder | Swedish-born | 543 (103.5) | **7.51 (6.80 to 8.29)** | **5.84 (5.24 to 6.51)** |
| | Non-refugee migrants | 10 (49.6) | **3.60 (1.93 to 6.71)** | **2.57 (1.38 to 4.81)** |
| | Refugees | 16 (77.0) | **5.60 (3.42 to 9.16)** | **3.76 (2.29 to 6.18)** |
| | | External causes | | |
| No mental disorder | Swedish born | 2912 (28.0) | 1 (REF) | 1 (REF) |
| | Non-refugee migrants | 98 (29.1) | 1.04 (0.85 to 1.27) | 0.85 (0.69 to 1.04) |
| | Refugees | 111 (24.9) | 0.89 (0.74 to 1.08) | **0.76 (0.62 to 0.91)** |
| Mental disorder | Swedish born | 961 (183.3) | **6.55 (6.08 to 7.04)** | **5.02 (4.64 to 5.43)** |
| | Non-refugee migrants | 25 (124.0) | **4.43 (2.99 to 6.57)** | **3.01 (2.03 to 4.48)** |
| | Refugees | 40 (192.5) | **6.88 (5.04 to 9.40)** | **4.37 (3.19 to 5.98)** |
| **Somatic disorder status** | | | | |
| | | All-cause mortality | | |
| No somatic disorder | Swedish born | 2003 (37.7) | 1 (REF) | 1 (REF) |
| | Non-refugee migrants | 55 (34.4) | 0.92 (0.70 to 1.20) | **0.71 (0.54 to 0.93)** |
| | Refugees | 74 (34.3) | 0.91 (0.72 to 1.15) | 0.82 (0.65 to 1.03) |
| Somatic disorder | Swedish born | 3831 (68.4) | **1.82 (1.72 to 1.92)** | **1.59 (1.51 to 1.68)** |
| | Non-refugee migrants | 131 (66.6) | **1.77 (1.48 to 2.11)** | **1.30 (1.08 to 1.55)** |
| | Refugees | 155 (62.0) | **1.65 (1.40 to 1.94)** | **1.25 (1.06 to 1.48)** |
| | | Suicide | | |
| No somatic disorder | Swedish born | 758 (14.3) | 1 (REF) | 1 (REF) |
| | Non-refugee migrants | 13 (8.1) | **0.57 (0.33 to 0.99)** | **0.47 (0.27 to 0.81)** |
| | Refugees | 21 (9.7) | 0.68 (0.44 to 1.05) | **0.63 (0.41 to 0.97)** |
| Somatic disorder | Swedish born | 1219 (21.8) | **1.53 (1.39 to 1.67)** | **1.28 (1.16 to 1.40)** |
| | Non-refugee migrants | 26 (13.2) | 0.93 (0.63 to 1.37) | 0.68 (0.46 to 1.01) |
| | Refugees | 39 (15.6) | 1.10 (0.79 to 1.51) | 0.81 (0.59 to 1.12) |
| | | External causes | | |
| No somatic disorder | Swedish born | 1461 (27.5) | 1 (REF) | 1 (REF) |
| | Non-refugee migrants | 36 (22.5) | 0.82 (0.59 to 1.15) | **0.64 (0.46 to 0.89)** |
| | Refugees | 51 (23.6) | 0.86 (0.65 to 1.14) | 0.77 (0.58 to 1.01) |
| Somatic disorder | Swedish born | 2412 (43.1) | **1.57 (1.47 to 1.67)** | **1.35 (1.26 to 1.44)** |
| | Non-refugee migrants | 87 (44.2) | **1.61 (1.30 to 2.00)** | 1.17 (0.94 to 1.45) |
| | Refugees | 100 (40.0) | **1.46 (1.19 to 1.78)** | 1.07 (0.87 to 1.31) |

Values in bold are statistically significant associations (p value<0.05).
HRs with 95% CI.
*Crude.
†Adjusted for age, sex, education, family situation, type of living area, unemployment, sickness absence and disability pension at baseline. For exposure groups related to mental disorders, model 2 was also adjusted for somatic morbidity in 2000–2004, and for exposure groups related to somatic disorders, model 2 was also adjusted for mental disorders in 2000–2004.

**Table 4** Associations between specific disorders (in where there are >10 cases in either refugees or non-refugee migrants) and all-cause mortality in Swedish-born individuals and refugee and non-refugee migrants, aged 16–25 years old residing in Sweden in 2000–2004

| Specific mental/somatic disorder status | Refugee status | n (rate per 100 000 person-years) | Model 1* | Model 2† |
|---|---|---|---|---|
| No substance misuse disorder | Swedish born | 5158 (48.0) | 1 (REF) | 1 (REF) |
| | Non-refugee migrants | 167 (47.6) | 1.00 (0.85 to 1.16) | **0.79 (0.68 to 0.93)** |
| | Refugees | 197 (42.8) | 0.89 (0.78 to 1.03) | **0.77 (0.67 to 0.89)** |
| Substance misuse disorder | Swedish born | 676 (397.8) | **8.30 (7.66 to 8.99)** | **5.14 (4.73 to 5.59)** |
| | Non-refugee migrants | 19 (307.0) | **6.41 (4.09 to 10.06)** | **3.48 (2.21 to 5.46)** |
| | Refugees | 32 (512.8) | **10.73 (7.58 to 15.18)** | **5.28 (3.73 to 7.49)** |
| No infectious and parasitic disease | Swedish born | 5418 (51.6) | 1 (REF) | 1 (REF) |
| | Non-refugee migrants | 168 (49.6) | 0.96 (0.83 to 1.12) | **0.77 (0.66 to 0.90)** |
| | Refugees | 215 (48.4) | 0.94 (0.82 to 1.08) | **0.82 (0.72 to 0.94)** |
| Infectious and parasitic disease | Swedish born | 416 (98.0) | **1.90 (1.72 to 2.10)** | **1.71 (1.55 to 1.90)** |
| | Non-refugee migrants | 18 (100.1) | **1.94 (1.22 to 3.09)** | 1.34 (0.84 to 2.12) |
| | Refugees | 14 (63.0) | 1.22 (0.72 to 2.06) | 0.90 (0.53 to 1.53) |
| No neoplasm | Swedish born | 5532 (52.2) | 1 (REF) | 1 (REF) |
| | Non-refugee migrants | 181 (51.8) | 1.00 (0.86 to 1.15) | **0.79 (0.68 to 0.92)** |
| | Refugees | 219 (48.0) | 0.92 (0.80 to 1.05) | **0.80 (0.70 to 0.92)** |
| Neoplasm | Swedish born | 302 (91.8) | **1.76 (1.57 to 1.97)** | **1.98 (1.76 to 2.22)** |
| | Non-refugee migrants | <10 | 1.31 (0.55 to 3.16) | 1.16 (0.48 to 2.78) |
| | Refugees | 40 (102.3) | **1.96 (1.06 to 3.65)** | **1.96 (1.05 to 3.64)** |
| No disease of the respiratory system | Swedish born | 5235 (51.4) | 1 (REF) | 1 (REF) |
| | Non-refugee migrants | 166 (49.6) | 0.97 (0.83 to 1.13) | **0.77 (0.66 to 0.90)** |
| | Refugees | 210 (48.0) | 0.94 (0.81 to 1.07) | **0.82 (0.71 to 0.94)** |
| Disease of the respiratory system | Swedish born | 599 (81.9) | **1.60 (1.47 to 1.74)** | **1.49 (1.37 to 1.62)** |
| | Non-refugee migrants | 20 (89.4) | **1.74 (1.12 to 2.70)** | 1.19 (0.77 to 1.85) |
| | Refugees | 19 (66.8) | 1.30 (0.83 to 2.04) | 0.98 (0.62 to 1.53) |
| No disease of the digestive system | Swedish born | 5342 (51.6) | 1 (REF) | 1 (REF) |
| | Non-refugee migrants | 173 (51.3) | 1.00 (0.86 to 1.16) | **0.79 (0.68 to 0.92)** |
| | Refugees | 207 (47.1) | 0.91 (0.80 to 1.05) | **0.80 (0.69 to 0.92)** |
| Disease of the digestive system | Swedish born | 492 (87.2) | **1.69 (1.54 to 1.85)** | **1.49 (1.36 to 1.63)** |
| | Non-refugee migrants | 13 (66.2) | 1.29 (0.75 to 2.22) | 0.88 (0.51 to 1.52) |
| | Refugees | 22 (82.7) | **1.61 (1.06 to 2.44)** | 1.16 (0.76 to 1.76) |
| No musculoskeletal disease | Swedish born | 5296 (51.9) | 1 (REF) | 1 (REF) |
| | Non-refugee migrants | 177 (53.0) | 1.02 (0.88 to 1.19) | **0.81 (0.70 to 0.94)** |
| | Refugees | 208 (47.6) | 0.92 (0.80 to 1.05) | **0.80 (0.70 to 0.92)** |
| Musculoskeletal disease | Swedish born | 538 (74.7) | **1.44 (1.32 to 1.57)** | **1.31 (1.20 to 1.43)** |
| | Non-refugee migrants | <10 | 0.76 (0.40 to 1.46) | 0.55 (0.29 to 1.06) |
| | Refugees | 21 (71.7) | 1.38 (0.90 to 2.12) | 1.00 (0.65 to 1.53) |
| No disease of the nervous system | Swedish born | 5387 (50.2) | 1 (REF) | 1 (REF) |
| | Non-refugee migrants | 170 (48.6) | 0.97 (0.83 to 1.13) | **0.78 (0.67 to 0.91)** |
| | Refugees | 215 (47.0) | 0.94 (0.82 to 1.07) | **0.82 (0.71 to 0.94)** |

**Table 4** Continued

| Specific mental/somatic disorder status | Refugee status | n (rate per 100 000 person-years) | Model 1* | Model 2† |
|---|---|---|---|---|
| Disease of the nervous system | Swedish born | 447 (231.0) | **4.60 (4.18 to 5.07)** | **2.83 (2.56 to 3.13)** |
| | Non-refugee migrants | 16 (236.0) | **4.71 (2.89 to 7.70)** | **2.45 (1.50 to 4.01)** |
| | Refugees | 14 (161.6) | **3.22 (1.91 to 5.44)** | **1.77 (1.05 to 3.00)** |

Values in bold are statistically significant associations (p value<0.05).
HRs with 95% CI.
*Crude.
†Adjusted for age, sex, education, family situation, type of living area, unemployment, sickness absence, disability pension at baseline and somatic disorders in 2000–2004.

also the healthy out-migration theory, often named the salmon bias (potential return-migration to the country of birth when migrants are very sick), and flaws in the registration of migrants may be other reasons for the mortality advantage among migrants.[18] We could now also show a mortality advantage in young migrants, despite an earlier report on a higher mortality risk in younger (particular children) versus older migrants.[18] This latter paper reported a U-shaped pattern of the relation between age and mortality risk in migrants. The explanation most in line with this observed pattern was an in-migration selection effect. Namely that children most often arriving with a caretaker might not be subject to health selection to the same extent as their adolescent and young adult counterparts. The U-shaped pattern might, therefore, be more explained by the composition of the young migrants than by pure age effects. In our study, approximately half of the young migrants arrived to Sweden more than 10 years before baseline (ie, during childhood) and the other half more recently. As our study population is young and we censored for emigration, salmon bias or faulty registration practices could be less likely explanations for the findings of mortality advantage among migrant youths over the Swedish-born youths in this study.

While estimates for all-cause mortality and mortality due to external causes were comparable, we found the suicide risk in the young migrants to be particularly low. This finding is in contrast to previous reports on an increased suicide risk in young asylum-seeking minors,[31] but in line with studies on refugees with a residence permit where no excess risk in suicide has been reported.[32] An explanation for the low suicide risk in young migrants with residence permit compared with those still facing an asylum-seeking process might be both the protective effect of receiving a residence permit and the culturally determined strong stigmatisation of suicide. Previous research from Denmark showed that long asylum decision-waiting periods were associated with an increased risk of psychiatric disorders.[33] Host countries are, therefore, advised to be aware of the mental health development of migrants during the asylum-seeking process.

Our study examined the associations between mental and somatic disorders and subsequent all-cause and cause-specific mortality in young refugees and non-refugee migrants. Studies investigating these associations in the general population—without a focus on migrants—have shown that individuals with mental disorders have a considerable higher risk for death due to external causes and suicide.[21 34] This is also true for somatic disorders but to a lesser extent.[34] In the present study, young Swedish-born individuals with a mental disorder had a sixfold and fivefold elevated risk for suicide and mortality due to external causes compared with their peers without a mental disorder, respectively. Also, the risk for preterm mortality was increased with 60% in Swedish born with specialised healthcare due to any somatic disorder compared with their counterparts without such healthcare. We can now further show that in the presence of a mental disorder, the risk of suicide and mortality due to external causes was lower in non-refugee migrants than in Swedish-born individuals. Similar patterns were found for somatic disorders, although with smaller risk differences. These findings are thought provoking and counterintuitive, as migrants with a mental or somatic disorder treated in specialised healthcare might be expected to suffer from disorders with a higher medical severity than their Swedish-born counterparts. This hypothesis arises from the common knowledge regarding discrepancies in access to and acceptance of specialised healthcare as well as treatment gaps of migrants compared with the population in high-income host countries.[24 35]

Two exceptions to this pattern occurred, however. Estimates for all-cause mortality were equally high for young refugees and Swedish born with substance use disorders and cancer.

These differences in patterns might be related to the high medical severity of these two disorders and the fact that culturally influenced resilience factors such as favourable health behaviour and strong social ties might be less relevant in protecting against preterm mortality related to these disorders.

### Methodological considerations
This study has several strengths, including the longitudinal population-based design and use of national registers with high completeness and validity. The large cohort size

allowed for detailed analyses of different types of mental and somatic disorders, the ability to adjust for important confounders and a rare outcome. Nevertheless, there are limitations. As information on primary healthcare was not available, we used data from specialised healthcare, which captures only the more severe forms of morbidity. Due to different barriers to access healthcare that are specific to immigrants,[36] we may have estimated differentially lower specialised healthcare use among refugees and non-refugee migrants than the Swedish born. This may lead to misclassification of undiagnosed and/or untreated individuals as unexposed. Potential differences in healthcare-seeking behaviour in Swedish-born individuals and migrants may also affect our findings. Moreover, there was a lack of data on important covariates, for example, social connectedness and health behaviour. Furthermore, the rare outcome may lead to lower statistical power for some of the analyses. Additionally, generalisability of our results is limited to countries with similar healthcare and social welfare systems, and where the ethnic composition of the immigrant population is not significantly different than our study population. Moreover, these results can only be generalised to resettled refugees and non-refugee migrants dwelling in the community rather than those living in refugee camps or immigration detention centres.

## CONCLUSIONS

The risk of all-cause mortality as well as specific mortality due to suicide and death due to external causes is lower among young refugees and non-refugee migrants compared with Swedish-born individuals. The lower rate of mortality of refugees can also be seen among those with specific mental and somatic disorders. With few exceptions, young migrants with specific mental and somatic disorders have a mortality advantage compared with their Swedish-born peers with the same disorders. Further research on protective factors is warranted.

**Author affiliations**
¹Department of Clinical Neuroscience, Division of Insurance Medicine, Karolinska Institutet, Stockholm, Sweden
²Department of Neuroscience, Uppsala University, Uppsala, Sweden
³Department of Psychosis Studies, King's College London, London, UK
⁴Karl-Landsteiner Institute for Health Promotion Research, Sitzenberg-Reidling, Austria
⁵Division of Insurance Medicine, Department of Clinical Neuroscience, Karolinska Institutet, Stockholm, Sweden

**Contributors** The study was designed by MH, EB and EM-R. EB conducted the data analyses. EM-R is responsible for the overall content as the guarantor. EM-R and MH drafted the manuscript. SF, SGR, AC, TD, KG and RA contributed with intellectual content and commented to results and successive drafts.

**Funding** The work was supported by the Swedish Research Council (grant number 2018-05783).

**Competing interests** None declared.

**Patient and public involvement** Patients and/or the public were not involved in the design, or conduct, or reporting, or dissemination plans of this research.

**Patient consent for publication** Not applicable.

**Ethics approval** Ethical approval of this study was given by the Regional Ethical Board in Stockholm (2007/762-31). The Ethical board also waived the requirement of informed consent from the participants in this study.

**Provenance and peer review** Not commissioned; externally peer reviewed.

**Data availability statement** No data are available. These data cannot be made publicly available due to privacy regulations. According to the General Data Protection Regulation, the Swedish law SFS 2018:218, the Swedish Data Protection Act, the Swedish Ethical Review Act and the Public Access to Information and Secrecy Act, these types of sensitive data can only be made available for specific purposes, including research, that meets the criteria for access to this type of sensitive and confidential data as determined by a legal review. Readers may contact Ellenor.Mittendorfer-Rutz regarding the data.

**ORCID iDs**
Magnus Helgesson http://orcid.org/0000-0002-7868-9712
Katalin Gémes http://orcid.org/0000-0003-3335-7850
Ridwanul Amin http://orcid.org/0000-0002-4491-3990
Ellenor Mittendorfer-Rutz http://orcid.org/0000-0001-5227-0721

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
