## [Reviewer comments · BMJ Open]

ARTICLE DETAILS

TITLE (PROVISIONAL)	Mental and somatic disorders and the subsequent risk of all-cause and cause-specific mortality in refugees, non-refugee migrants and the Swedish-born youth. A population-based cohort study in Sweden
AUTHORS	Helgesson, Magnus; Björkenstam, Emma; Filatova, Svetlana; Rahman, Syed; Cullen, Alexis; Dorner, Thomas; Gémes, Katalin; Amin, Ridwanul; Mittendorfer-Rutz, Ellenor

VERSION 1 – REVIEW

REVIEWER	Natasha Saunders Hospital for Sick Children, Pediatrics
REVIEW RETURNED	06-Oct-2021

GENERAL COMMENTS	The use of the terms mental, somatic, suicide and external causes are somewhat confusing and perhaps not as familiar to the readership. In particular, external causes may include suicide (this is typical in the injury literature) – could this be called something different (e.g. unintentional injuries?). Title: Please add study type/design. Abstract: The abstract in general needs much more detail in the methods and results. (methods) – It is not clear your study design. Your main outcome measures are not specified, nor are the exposure groups. (results) – Need to add some basic descriptive findings rather than relative findings first. Please supply numbers, rates, mean follow up time, etc. Results need to show HR with CI's. For all findings in results, please attach numbers to them rather than just describing higher or lower risk. Introduction – lines 22-24 – Many studies have shown lower risk of mental disorders for refugees compared to the host population. Please better support this statement and ensure current literature reviewed to best support this statement (or alternatively refute it). Only a single study was cited here though the authors say, “studies”. The concept of “somatic disorders” (or perhaps just the terminology) is not typical and is not introduced until later in the introduction. Perhaps earlier on in the introduction, its definition could be included. Page 7 lines 12-19 – It is not clear to me what this sentence means (Differences of the relative importance...). Can this be clarified? The Introduction has a number of points that might be better suited for discussion and appear explanatory for expected findings, rather than provide background. I think this makes it harder to interpret/read the flow of the paper. The last paragraph (aims) – what is meant by “subsequent”? Subsequent to what? The aims are unclear – The authors mention all-cause and cause-specific mortality but then go on to say “general and related to
---

	specific mental and somatic disorders". Is this a duplicate concept or something different? This is confusing and it is not clear what the objectives are. Methods: In defining those with mental disorders, inpatient or specialist outpatient healthcare is used. However, most mental health care for conditions such as anxiety and depression (including severe or chronic disease) are managed in primary (not specialized) care. Can you justify the use of specialist care only to define this measure? What limitations does it have? Can you also clarify what is meant by specialist care (or how this is defined in your setting)? Results: On page 12, in presenting the HRs, please present the CI's too. Table 3 – It is not clear/obvious what the differences are between the top half and the bottom half of this table. Suggest adding a title to the rows to differentiate. Table 4 – Why are there numbers crossed off where n<10? Should these be deleted? Discussion: Main findings – There are multiple typos, please review carefully. Remove statement of precedent (i.e. this is the first study), rather, state the gap it fills. Please also remove it from page 18, lines 8-11. Some discussion about the generalizability of the findings to other jurisdictions outside of Europe would be helpful (especially to places like Canada and Australia where there are very high proportions of foreign born, including refugees). General: Consider using the RECORD Guideline for Reporting of observational studies using routinely collected data, rather than the STROBE checklist. https://www.record-statement.org/
--	--

REVIEWER	Ina Heiberg Center for Clinical Documentation and Evaluation
REVIEW RETURNED	07-Dec-2021

GENERAL COMMENTS	Review bmjopen-2021-054351 I have read this manuscript with interest. The article investigates the association between specific mental and somatic disorder and subsequent mortality among young refugees and non-refugee migrants compared to Swedish-born persons of the same age. This is an important topic and more research is needed. The data material which the article builds on is solid and the results are interesting. A particular strength is the population-based design, using national registers with high completeness. Overall, I find the manuscript tidy and easy to read, although with a number of typos and punctuation errors. My main comments are related to possible over-adjustment and the handling of time-dependent variables. I also have some questions about the exclusion criteria used when identifying immigrants. In addition to sex and age, the analyses are adjusted for education level, type of living area, family situation, sickness absence, disability pension and morbidity. All these latter covariates are measured at baseline, when the participants were aged 16-25 years old. First; did you considered the possibility that adjustment for such variables may introduce collider bias? Did you use directed acyclic graphs to assist in the selection of covariates? If not, could you please explain how you have reasoned about the inclusion of
--

	covariates? In the case of no collider bias, measurement of time-dependent variables (such as education level, type of living area, family situation, sickness absence, disability pension and morbidity) only at baseline may not be the best way of analysing this topic. The participants are in an age where such circumstances are particularly changeable. The majority of participants were younger than 20 at inclusion and many changes may occur during the 12-year follow up period (e.g. a number of chronic conditions may be diagnosed during the observation period, which are not captured in the analyses). I believe that the time dimension has to be addressed more thoroughly. For the covariates that are time-dependent I believe that there are alternative ways to operationalize these. Most importantly, these could be assessed as time-dependent covariates or repeated measurements (e.g. each year). I also have some questions about the selection of the study population. About one third of immigrants were excluded due to missing information on the reason for settlement. However, the results were not very different between refugees and non-refugee migrants. Did you investigate whether inclusion of those who had missing information on the reason for settlement could have changed results in any way? Also, 22% of non-refugee migrants were excluded as they did not come from the same countries as the refugees. Did you investigate how this may have affected the results? There are also some minor areas which would benefit from clarification. Questions about these are listed below. Introduction: You refer to data artefacts that might explain lower premature mortality among refugees (page 5, refs 10, 12-13). What are these artefacts? Are they relevant for this study, and could they be corrected? Study population Does the definition of refugees/migrants include those who do not yet have a permanent personal identity number? It would be interesting to know whether second generation immigrants also have a reduced mortality risk. Is it possible to identify those in your data? Covariate definition There is an inconsistency in the listing of covariates in the “covariates” section compared to the “Statistical analyses” section (morbidity is included in the latter section, but not the first). Is there any particular reason why diabetes, as the only condition, is omitted from the definition of somatic diseases (page 8)? Outcome measures Are emigration statistics reliable? (That is; do you know whether refugees/migrants actually reside in Sweden, and thus are included in national registries such as the National Patient Registry?) Does the Cause of Death Registry include deaths that occur among residents of Sweden when they are abroad? Statistical method
--	---

	Please explain in the methods section how the Cox proportional hazard assumptions are respected. How were age handled in these analyses? The variable “Family situation” seem to have a category that is based on age. How does this relate to the age variable? Did you investigate interaction effects? Results The unadjusted analyses show no difference in all-cause mortality and mortality from external causes between young refugees, young non-refugee migrants and Swedish-born persons of the same age. Which covariates contribute the most to changing the unadjusted results? Is there also a difference in mortality when only taking into account age, gender and country background? I think a more thorough discussion of what actually contributes to the changed conclusion is warranted. Which countries were the main contributors to the “Others” group in Table 1? They make up more than a third, and may need to be described somewhat better. Some of the tables are “heavy” to read. Could some of this be presented graphically in order to increase availability for the reader? Discussion Contradicting with the fact you mention in your introduction that immigrant populations have a higher health and socio-economic disadvantage, you find lower premature mortality risk in the immigrant groups. This surprising fact should be discussed in more detail. Does any of your included covariates offer any hypothetical explanation for this finding? The “salmon bias” is mentioned as a possible explanation for the lower mortality observed among refugees/migrants (page 16). Do you consider this to be a relevant explanation in this young age group? If referred to, the term may also need an explanation, as the readers of a general medical journal may not be familiar with the term.
--	---

VERSION 1 – AUTHOR RESPONSE

Reviewer 1

R1.1

The use of the terms mental, somatic, suicide and external causes are somewhat confusing and perhaps not as familiar to the readership. In particular, external causes may include suicide (this is typical in the injury literature) – could this be called something different (e.g. unintentional injuries?).

Response: Thank you very much for reviewing the manuscript and for the constructive feedback! We have included an additional explanation for external causes in the revised manuscript. As it includes “suicide codes” – an intentional act - it can not be called “unintentional injuries”. See changes in the methods part on page 9, line 21-22.

R1.2

Title: Please add study type/design.

Response: This was also requested by the Editor, as noted above the revised title is now called ‘Mental and somatic disorders and the subsequent risk of all-cause and cause-specific

mortality in young adults – are there differences between refugees, non-refugee migrants and the Swedish-born? A population-based cohort study in Sweden'

R1.3

Abstract: The abstract in general needs much more detail in the methods and results.

(methods) – It is not clear your study design. Your main outcome measures are not specified, nor are the exposure groups.

Response: Thank you for this helpful suggestion. We have revised the methods section in the following manner:

Methods: ~~The study population of this register study comprised all 1,003,760 individuals (including 40,305 refugees and 31,687 non-refugee migrants), 16-25 years old, residing in Sweden on 31 December 2004. Crude and multivariate Hazard ratios (HR) with 95% Confidence intervals (CI) were yielded by performing Cox regression models with a follow-up from 2005 to 2016.~~ In this register-based cohort study, all 1,003,760 individuals (40,305 refugees, 31,687 non-refugee migrants as the exposure groups and the rest as the Swedish-born comparison group), 16-25 years old, residing in Sweden on 31-Dec-2004 were included. These individuals were followed regarding the outcome of all-cause and cause-specific mortality (suicide and external causes) between 2005-2016. The study population was also stratified according to any use of specialised healthcare for mental or somatic diagnoses before baseline (2000-2004). Cox regression models yielding crude and multivariate hazard ratios (HR and aHR, respectively) with 95% confidence intervals (CI) were used to investigate the aforementioned associations.

R1.4

(results) – Need to add some basic descriptive findings rather than relative findings first. Please supply numbers, rates, mean follow up time, etc. Results need to show HR with CI's. For all findings in results, please attach numbers to them rather than just describing higher or lower risk.

Response: We have revised the results section in the abstract as much as possible. However, we could not add numbers for all results because the journal requires the abstract to be maximum 300 words. The revised text is as the following:

Results: A lower proportion of both refugees (12%) and non-refugee migrants (10%) had college/university education compared to the Swedish-born individuals (17%). The proportion of unemployed (>180 days) among refugees (2.3%) and non-refugees (2.9%) was higher than the Swedish-born (1.4%). Refugees and non-refugee migrants had about a 20% percent lower risk of all-cause mortality and mortality due to external causes compared to Swedish-born individuals. An even greater reduction in suicide risk (aHR 0.51, 95% CI 0.37-0.70, and 0.63, 95% CI 0.49-0.82 for non-refugees and refugees, respectively) was found.

R1.5

Introduction – lines 22-24 – Many studies have shown lower risk of mental disorders for refugees compared to the host population. Please better support this statement and ensure current literature

reviewed to best support this statement (or alternatively refute it). Only a single study was cited here though the authors say, “studies”.

Response: We have cited the following studies that support this statement:

“For these reasons, studies report a higher risk for mental disorders, particularly post-traumatic stress disorders in young refugees compared to the host population.⁶⁻¹²”

References:

6. Björkenstam E, Helgesson M, Norredam M, et al. Differences in psychiatric care utilization between refugees, non-refugee migrants and Swedish-born youth. *Psychol Med* 2020;1-11. doi: 10.1017/s0033291720003190 [published Online First: 2020/09/12]
7. Fazel M, Reed RV, Panter-Brick C, et al. Mental health of displaced and refugee children resettled in high-income countries: risk and protective factors. *The Lancet* 2012;379(9812):266-82. doi: 10.1016/s0140-6736(11)60051-2
8. Fazel M. Psychological and psychosocial interventions for refugee children resettled in high-income countries. *Epidemiology and psychiatric sciences* 2018;27(2):117-23. doi: 10.1017/S2045796017000695 [published Online First: 2017/11/10]
9. Norredam M, Nellums L, Nielsen RS, et al. Incidence of psychiatric disorders among accompanied and unaccompanied asylum-seeking children in Denmark: a nation-wide register-based cohort study. *European Child & Adolescent Psychiatry* 2018;27(4):439-46. doi: 10.1007/s00787-018-1122-3
10. Kien C, Sommer I, Faustmann A, et al. Prevalence of mental disorders in young refugees and asylum seekers in European Countries: a systematic review. *European Child & Adolescent Psychiatry* 2019;28(10):1295-310. doi: 10.1007/s00787-018-1215-z
11. Montgomery E. Trauma, exile and mental health in young refugees. *Acta psychiatrica Scandinavica* 2011;124(s440):1-46. doi: <https://doi.org/10.1111/j.1600-0447.2011.01740.x>
12. Manhica H, Almquist Y, Rostila M, et al. The use of psychiatric services by young adults who came to Sweden as teenage refugees: a national cohort study. *Epidemiology and psychiatric sciences* 2017;26(5):526-34. doi: 10.1017/s2045796016000445 [published Online First: 2016/06/30]

R1.6

The concept of “somatic disorders” (or perhaps just the terminology) is not typical and is not introduced until later in the introduction. Perhaps earlier on in the introduction, its definition could be included.

Response: We have now clarified this term the first time it appears in the introduction as ‘any non-psychiatric disorder’ (page 6, line 2).

R1.7

Page 7 lines 12-19 – It is not clear to me what this sentence means (Differences of the relative importance...). Can this be clarified?

Response: We have clarified this sentence in the following manner.

~~Differences of the relative importance of specific mental and somatic disorders for subsequent mortality risk in young refugees compared to young individuals in the host population are likely, but~~

~~are lacking to date.~~ Therefore, specific mental and somatic disorders may play a differential role for the mortality risk among young refugees compared to young individuals in the host population. However, such knowledge is lacking to date.

R1.8

The Introduction has a number of points that might be better suited for discussion and appear explanatory for expected findings, rather than provide background. I think this makes it harder to interpret/read the flow of the paper.

Response: Thank you, we have shortened the text according to your suggestions. However, some of these points are the rationale for the study, and therefore, we have kept them in the background and we have discussed these points in the discussion section as well. The following changes were made in the manuscript:

On page 5, lines 18-19:

~~Explanatory models include not only in- and out-migration health selection, but also cultural features such as better health behaviour but also data artefacts.~~

On page 6, lines 16-20:

~~There seem to be Studies report differences in the prevalence of both mental and somatic disorders as well as multimorbidity between subgroups of migrants. A study from Sweden found that the risk of a mental disorder is higher in refugee migrants compared to non-refugee migrants.¹⁹ Differences in the occurrence of multimorbidity between refugees, non-refugee migrants and the host population were also reported by a study from Norway.²⁰~~

R1.9

The last paragraph (aims) – what is meant by “subsequent”? Subsequent to what?

Response: We agree that this sentence is unclear, and have therefore amended as follows:

This study aims to address the mentioned literature gaps by investigating if the risk of ~~subsequent~~-all-cause and cause-specific mortality differs in young refugees and non-refugee migrants compared to individuals in the host population of the same age.

R1.10

The aims are unclear – The authors mention all-cause and cause-specific mortality but then go on to say “general and related to specific mental and somatic disorders”. Is this a duplicate concept or something different? This is confusing and it is not clear what the objectives are.

Response: We are grateful to the reviewer for pointing out that the aims were unclear, the revised aims are as follows:

“This study aims to address the aforementioned gaps in the literature by investigating if the risk of all-cause and cause-specific mortality differs in young refugees and non-refugee migrants compared to individuals in the host population of the same age. A further aim was to investigate if the potential differences in all-cause and cause-specific mortality vary by a prior diagnosis of specific mental and somatic disorders.”

R1.11

Methods:

In defining those with mental disorders, inpatient or specialist outpatient healthcare is used. However, most mental health care for conditions such as anxiety and depression (including severe or chronic disease) are managed in primary (not specialized) care. Can you justify the use of specialist care only to define this measure? What limitations does it have? Can you also clarify what is meant by specialist care (or how this is defined in your setting)?

Response: Whilst we agree that this is a limitation, unfortunately, primary healthcare is not available for entire Sweden, only for the Stockholm County. Because refugees and non-refugee migrants are minorities in Sweden, we wanted to include the entire population to ensure adequate statistical power in our analyses. One limitation of using data from specialised healthcare (hospitalisations and specialised outpatient healthcare) is that, as you have mentioned, we only capture the more severe forms of morbidity. Due to different barriers to access healthcare that are specific to migrants, we may have captured a lower proportion of refugees and non-refugee migrants who used specialised healthcare at baseline than the Swedish-born. This issue is now discussed in detail the limitation section (page 19, lines 23-26, page 20, lines 1-3). The term ‘specialised healthcare’ is now clarified on page 8, line 23. The same information was removed from page 9, line 4 to avoid repetition.

R1.12

Results:

On page 12, in presenting the HRs, please present the CI's too.

Response: We have added the CIs.

R1.13

Table 3 – It is not clear/obvious what the differences are between the top half and the bottom half of this table. Suggest adding a title to the rows to differentiate.

Response: Thank you for this helpful suggestion to make this table clearer, we have now amended accordingly.

R1.14

Table 4 – Why are there numbers crossed off where $n < 10$? Should these be deleted?

Response: This was a typographical error, we have removed the ‘strikethrough’ formatting.

R1.15

Discussion:

Main findings – There are multiple typos, please review carefully.

Response: Thank you for identifying these issues, which we have now corrected.

R1.16

Remove statement of precedent (i.e. this is the first study), rather, state the gap it fills. Please also remove it from page 18, lines 8-11.

Response: We have revised these sentences accordingly (page 16, line 3, and page 17, line 19).

R1.17

Some discussion about the generalizability of the findings to other jurisdictions outside of Europe would be helpful (especially to places like Canada and Australia where there are very high proportions of foreign born, including refugees).

Response: Thank you for identifying this important issue, we have revised the Discussion as follows page 19, lines 9-14.

R1.18

General:

Consider using the RECORD Guideline for Reporting of observational studies using routinely collected data, rather than the STROBE checklist. <https://www.record-statement.org/>

Response: We have added the RECORD checklist and amended the manuscript accordingly.

The following changes were made:

- **The title is updated**
- **Page 8, lines 3, 11-13, 16**
- **Sensitivity analyses??**

Reviewer 2

R2.1

I have read this manuscript with interest. The article investigates the association between specific mental and somatic disorder and subsequent mortality among young refugees and non-refugee migrants compared to Swedish-born persons of the same age. This is an important topic and more

research is needed.

The data material which the article builds on is solid and the results are interesting. A particular strength is the population-based design, using national registers with high completeness.

Overall, I find the manuscript tidy and easy to read, although with a number of typos and punctuation errors.

My main comments are related to possible over-adjustment and the handling of time-dependent variables. I also have some questions about the exclusion criteria used when identifying immigrants. In addition to sex and age, the analyses are adjusted for education level, type of living area, family situation, sickness absence, disability pension and morbidity. All these latter covariates are measured at baseline, when the participants were aged 16-25 years old.

Response: We thank the reviewer for their comments and valuable suggestions for improvement which we respond to below.

R2.2

First; did you considered the possibility that adjustment for such variables may introduce collider bias? Did you use directed acyclic graphs to assist in the selection of covariates? If not, could you please explain how you have reasoned about the inclusion of covariates?

Response: We are grateful to the reviewer for this helpful suggestion. The directed acyclic graph (DAG) did not reveal any biased pathway due to colliders. The DAG codes are provided at the end of this file and can be replicated at <http://www.dagitty.net/dags.html>. We did not intend to specify the best parsimonious model that explains the association rather wanted to include all covariates that are, in theory and according to the literature, associated with both the exposure and the outcome but do not fall in the causal pathway, and for which data was available.

R2.3

In the case of no collider bias, measurement of time-dependent variables (such as education level, type of living area, family situation, sickness absence, disability pension and morbidity) only at baseline may not be the best way of analysing this topic. The participants are in an age where such circumstances are particularly changeable. The majority of participants were younger than 20 at inclusion and many changes may occur during the 12-year follow up period (e.g. a number of chronic conditions may be diagnosed during the observation period, which are not captured in the analyses). I believe that the time dimension has to be addressed more thoroughly. For the covariates that are time-dependent, I believe that there are alternative ways to operationalize these. Most importantly, these could be assessed as time-dependent covariates or repeated measurements (e.g. each year).

Response: We thank the reviewer for this important comment. While introduction of time-dependent variables would have been an elegant way of dealing with changing conditions during follow-up, we regret that we don't have access to these variables in the dataset.

R2.4

I also have some questions about the selection of the study population. About one-third of immigrants were excluded due to missing information on the reason for settlement. However, the results were not very different between refugees and non-refugee migrants. Did you investigate whether inclusion of those who had missing information on the reason for settlement could have changed results in any way?

Response: The reviewer raises an important point, these immigrants were excluded because we could not determine whether they were refugees or non-refugee immigrants because this distinction depends on the availability of the information on reason for resettlement.

According to Statistics Sweden (SCB), individuals with incomplete information regarding this variable are primarily EU citizens who do not register their (and their families') right of residence with the Swedish Tax Agency (Statistics Sweden, 2011). As the results for refugees and non-refugee migrants were not very different, the inclusion of the migrants without information on reason for settlement would not have changed the results.

Reference:

STATISTICS SWEDEN (SCB). 2011. Review of previously published statistics regarding reason for residence (In Swedish). [Cited 25 December 2021]. Available from <https://www.scb.se/contentassets/9171f415739b4211addb78298247d3bc/oversyn-av-tidigare-publicerad-statistik-grund-for-bosattning.pdf>.

R2.5

Also, 22% of non-refugee migrants were excluded as they did not come from the same countries as the refugees. Did you investigate how this may have affected the results?

Response: Our aim was to include non-refugee immigrants from the same country of birth as the refugees to test the hypothesis that if traumatic experience as a refugee had affected mortality outcomes given that the post-migration conditions and the cultural background can be considered similar between these migrant subgroups. Therefore, we did not include non-refugee immigrants from other countries than the refugees. Further studies are warranted to investigate risk estimates for non-refugee migrants from these countries.

R2.6

There are also some minor areas which would benefit from clarification. Questions about these are listed below.

Introduction:

You refer to data artefacts that might explain lower premature mortality among refugees (page 5, refs 10, 12-13). What are these artefacts? Are they relevant for this study, and could they be corrected?

Response: Wallace and Kulu (2014), in their analysis of migrant mortality in England and Wales, summarised forms of data artifact that may lead to the presumably false conclusion that migrants are healthier than the general population: (1) age can be misreported; (2)

nationality and/or ethnicity may be misclassified; (3) different forms of return migration may lead to numerator-denominator error, causing returned migrants to indefinitely age in population registers. Given that the study population is young in this study, and we censored for emigration, we believe that these issues have not (or minimally) affected our results. We have still added some information in the manuscript.

Reference:

Wallace, M., & Kulu, H. (2014). Low immigrant mortality in England and Wales: A data artefact?. Social Science & Medicine, 120, 100–109.

R2.7

Study population

Does the definition of refugees/migrants include those who do not yet have a permanent personal identity number?

Response: We can confirm that temporary residents for <365 days, asylum seekers and undocumented migrants do not have a personal identity number in Sweden, and therefore, their information is not available in the administrative registers. We agree that this is important information that should be clarified in the manuscript, we have therefore amended the Methods section as follows:

On page 8, line 11-15:

Temporary residents for <365 days, asylum seekers and undocumented migrants do not have a personal identity number in Sweden, and therefore, their information is not available in the administrative registers.

R2.8

It would be interesting to know whether second generation immigrants also have a reduced mortality risk. Is it possible to identify those in your data?

Response: We agree that this is a very interesting research question. Still, this research question was outside the scope of this study and we feel that we already now have an abundance of results.

R2.9

Covariate definition

There is an inconsistency in the listing of covariates in the “covariates” section compared to the “Statistical analyses” section (morbidity is included in the latter section, but not the first).

Is there any particular reason why diabetes, as the only condition, is omitted from the definition of somatic diseases (page 8)?

Response: We regret that this typo created the confusion. Diabetes Mellitus (ICD-10: E10-E14) was included as other somatic disorders. This information is now corrected (page 9, line 17).

R2.10

Outcome measures

Are emigration statistics reliable? (That is; do you know whether refugees/migrants actually reside in Sweden, and thus are included in national registries such as the National Patient Registry?)

Response: We are confident that the emigration statistics are reliable. All individuals who emigrate are expected to notify the tax agency and this information is updated in all administrative registers. As it is possible that not all individuals will report this information as required, we additionally identified all individuals who did not have any information in the population register for two consecutive years and considered that they have emigrated during the former year (if they had not died before that). This way we could compensate for the emigrations that were not notified to the tax agency.

R2.11

Does the Cause of Death Registry include deaths that occur among residents of Sweden when they are abroad?

Response: Yes, from 2012, this register includes deaths that occur among residents of Sweden when they are abroad. This information has been added to the manuscript on page 9, lines 1-2.

Reference:

Brooke HL, Talbäck M, Hörnblad J, Johansson LA, Ludvigsson JF, Druid H, Feychting M, Ljung R. The Swedish cause of death register. Eur J Epidemiol. 2017 Sep;32(9):765-773. doi: 10.1007/s10654-017-0316-1.

R2.12

Statistical method

Please explain in the methods section how the Cox proportional hazard assumptions are respected. How were age handled in these analyses?

Response: The following information are added in the statistical analyses section: “The proportional hazard assumption was verified graphically using log-minus-log plot (plotting the log-log transformation of the survival function against survival time) over the different categories for each of the covariates and exposure variables.” (on page 10, lines 8-10) Calendar day was used as the underlying time scale and age at study entry was adjusted for in the multivariate-adjusted models. (on page 10, lines 12-13).

R2.13

The variable “Family situation” seem to have a category that is based on age. How does this relate to the age variable?

Response: There is a relation between the variables family situation and age. Still, we decided to include both. Family situation, however, had only a marginal influence in the multivariate analysis.

R2.14

Did you investigate interaction effects?

Response: The interaction between refugee status and mental/somatic disorders is shown in stratified analyses. Analyses of additional interactions were not the scope of this study.

R2.15

Results

The unadjusted analyses show no difference in all-cause mortality and mortality from external causes between young refugees, young non-refugee migrants and Swedish-born persons of the same age. Which covariates contribute the most to changing the unadjusted results? Is there also a difference in mortality when only taking into account age, gender and country background? I think a more thorough discussion of what actually contributes to the changed conclusion is warranted.

Response: Thank you for this interesting question. The estimates in the multivariate models in these analyses were only to a minor extent altered by adjustment for the various covariates. The variable which contributed most to the changed estimates was – non surprisingly – age. Other variables only contributed marginally. Due to the lack of any important information in the adjusted analyses, we believe that a thorough discussion of the effect on age is not necessary given the well-known increasing risk of mortality with age.

R2.16

Which countries were the main contributors to the “Others” group in Table 1? They make up more than a third, and may need to be described somewhat better.

Response: We have updated the categories in Table 1 and rearranged the rows to improve clarity. The categories for country/region of birth are: Horn of Africa, Afghanistan, Iran, Iraq, Syria, Other countries in Asia (except Afghanistan, Iran, Iraq and Syria), former Yugoslavian countries, Other countries (except Horn of Africa, Asian countries, former Yugoslavian countries) include mainly countries in South America and the remaining African countries.

R2.17

Some of the tables are “heavy” to read. Could some of this be presented graphically in order to increase availability for the reader?

Response: We believe it will be challenging to present so many HRs with CIs graphically. To improve clarity, we have rearranged the column 1 and 2 in Table 3 so that the HRs for those with or without mental or somatic disorders, stratified by refugee status, can be compared more easily. Moreover, we have marked the statistically significant estimates in bold and added a footnote on this change. The same steps were done in Table 4.

R2.18

Discussion

Contradicting with the fact you mention in your introduction that immigrant populations have a

higher health and socio-economic disadvantage, you find lower premature mortality risk in the immigrant groups. This surprising fact should be discussed in more detail. Does any of your included covariates offer any hypothetical explanation for this finding?

Response: Thank you for this important comment. This is an often found result, and the covariates do not seem to explain this. The main reason for this association lies in the selection effects prior to having a residence permit. We discuss this in the discussion part.

R2.19

The “salmon bias” is mentioned as a possible explanation for the lower mortality observed among refugees/migrants (page 16). Do you consider this to be a relevant explanation in this young age group? If referred to, the term may also need an explanation, as the readers of a general medical journal may not be familiar with the term.

Response: The reviewer raises an important point, we agree that salmon bias could be less likely explanations for the findings of mortality advantage among migrant youths over the Swedish-born youths in this study provided that our study population is comparatively younger, and we censored for emigration. This information is added on page 18, lines 9-12. We have also added an explanation of the salmon bias on page 17, lines 23-24.